# A Novel Genotype of *Orientia tsutsugamushi* in Human Cases of Scrub Typhus from Southeastern India

**DOI:** 10.3390/microorganisms13020333

**Published:** 2025-02-04

**Authors:** Krishnamoorthy Nallan, Bhuvaneshwari Chinnathambi Kalidoss, Eunice Swarna Jacob, Samyuktha Krishnasamy Mahadevan, Steny Joseph, Ramkumar Ramalingam, Govindarajan Renu, Balaji Thirupathi, Balajinathan Ramasamy, Bhavna Gupta, Manju Rahi, Paramasivan Rajaiah

**Affiliations:** 1ICMR-Vector Control Research Centre, Field Unit, 4, Sarojini Street, Madurai 625002, India; stenyjosephv@gmail.com (S.J.); rammedmicro@gmail.com (R.R.); govindarajan.r@icmr.gov.in (G.R.); balaji.t@icmr.gov.in (B.T.); bhavna.g@icmr.gov.in (B.G.); 2Department of Microbiology, Government Medical College and Hospital, Thanjavur 613004, India; rsiddn1967@gmail.com (B.C.K.); swarnakumar369@gmail.com (E.S.J.); drbalajinatha@yahoo.co.in (B.R.); 3Department of Biotechnology, Stella Maris College, Chennai 600086, India; samyukthamahadeven@gmail.com; 4ICMR-Vector Control Research Centre, Puducherry 625006, India; drmanju.r@icmr.gov.in

**Keywords:** GroEL, 56 kDA, genotypes, genetic heterogeneity, scrub typhus, *Orientia tsutsugamushi*, chigger, Karp, Kato, Kawasaki, India

## Abstract

Scrub typhus is a mite-borne, re-emerging public health problem in India, particularly in Tamil Nadu, South India. More than 40 serotypes of *Orientia tsutsugamushi* have been documented worldwide. However, the information on the circulation of its molecular sub-types in India is scanty. A retrospective study was conducted among serologically confirmed cases of scrub typhus. DNA isolated from blood was screened by a nested polymerase chain reaction (nPCR) targeting the GroEL and the 56 kDa type-specific antigen (TSA) genes. Out of 59 samples, 14 partial fragments of GroEL and the twelve 56 kDa genes were PCR-amplified and DNA-sequenced. The neighbor-joining (NJ) analysis indicated three distinct phylogenetic clades, including a novel genotype designated as *Ot*-Thanjavur-Tamil Nadu (*Ot*-TJTN, 9 nos. 64.3%); Karp-like (4 nos. 28.6%); and Kuroki-Gilliam type (1 no. 7.1%). Also, phylogenetic analysis of twelve 56 kDa variable domains (VDΙ-ΙΙΙ) of TSA gene sequences revealed a distinctive new genotypic cluster of eight samples (66.6%), and the remaining four (33.4%) were Karp-like genotypes. The Simplot analysis for the similarity and event of recombination testing elucidated the existence of the new genotype of the *Ot*-TJTN cluster, which was undescribed so far, in the Kato and TA716 lineages. The significant findings recommend further studies to understand the ongoing transmission dynamics of different *O. tsutsugamushi* strains in vector mites, rodent hosts, and humans in this region.

## 1. Introduction

Scrub typhus (St), a re-merging mite-borne infection of humans, has a widespread distribution in Southeast Asia (SEA) [1,2,3]. *Orientia tsutsugamushi* (*Ot*), the causative agent of St, has been found to exist as multiple genotypes [4] and remain in circulation among rodents and mite vectors in nature [5,6,7]. The incriminated primary mite vectors are classified under the genera *Leptotrombidium* spp., *Schoengastiella* spp., and *Ornithonyssus* spp. [6,8,9]. The disease is known to be endemic mainly in the “Tsutsugamushi Triangle”, i.e., in Southeast Asian countries [10]; it is estimated that around one billion people are suffering from scrub typhus fever, and one million people are affected [11]. Scrub typhus is considered one of the leading infectious diseases in the Asia–Pacific region [12]. In India, the case fatality ratio of scrub typhus ranged from 12 to 623 during the years 2002–2014, and St cases have been widely distributed in many states in the country, particularly in Himachal Pradesh, Assam, and Tamil Nadu [3,13]. Scrub typhus is misdiagnosed and underreported because of its overlapping signs and symptoms with the other commonly occurring leptospirosis, dengue, brucellosis, typhoid, etc. [12,14,15]. Though the causative agent was believed to exist as a single species (*Orientia tsutsugamushi*), the recent discovery of two new species (Ca. O. chuto sp. nov. and Ca. O. chiloensis) has indicated its rich diversity [16,17]. Globally, more than 40 antigenic subtypes of *Ot* have been documented [18].

Though St is a re-emerging zoonotic problem in India [19,20,21], the genetic diversity of the circulating pathogen has not been determined in endemic areas [22]. Even though scrub typhus is considered a seriously emerging public health problem in this area (the proportion of cases is 37.6%) [20], the information available on the circulating genotypes, especially in the southeastern part of the country, is scanty. The Kato, Karp-like, Gilliam, Ikeda, Neimeng-65, TA716, and TA763 genotypes have been identified from different states in India [4,23,24,25]. However, routine clinical cases are treated based on the ELISA tests, which include the 56 kD outer membrane recombinant protein of Karp, Kato, Gilliam, and TA716 genotypes in the diagnostic panel [26].

*O. tsutsugamushi* has adapted to survive naturally in the mite population through vertical and horizontal transmission routes in the environment [27]. Since the different strains of *O. tsutsugamushi* co-exist in the mites, the chances of occurrence of genetic recombination between different strains and the creation of new subtypes are higher [28]. These subtypes may act as more virulent and develop clinical complications in humans [24]. Hence, an in-depth understanding of circulating genotypic study with the appropriate markers is essential. Among the different molecular markers, the GroEL and 56 kDa are more accurate in the detection of genotypes [27,29]. Therefore, in the present attempt, the GroEL and 56 kDa genes were used for genotyping of the *Ot* in this area.

Scrub typhus fever cases have been routinely diagnosed in patients seeking medical assistance at Government Thanjavur Medical College Hospital, located on the western edge of Tamil Nadu State. Due to the lack of a PCR-based molecular diagnostic study, there is no information available on the genotype of *O. tsutsugamushi* from this area. The GroEL gene-based earlier study with a limited number of human and rodent samples in Theni district, South India, revealed the occurrence of three genotypes [30]. Therefore, the present study aims to explore the presence of *Ot* genotypes from the Thanjavur district and compare the result with the genotypes detected in our previous studies by analyzing the GroEL and 56 kDa TSA protein genes.

## 2. Materials and Methods

### 2.1. Study Area

Thanjavur district lies on the east coast of Tamil Nadu between 78°45′ and 70°25′ E; 9°50′ and 11°25′ N in the delta, also called the “rice bowl” of Tamil Nadu. The inhabitants are involved in agricultural activities—mainly paddy cultivation [31], the rodent population is abundant, and different species of rodents (*R. rattus*, *B. bengalensis*, *S. murinus*, *T. indica*, and *M. musculus*) were recorded with chigger mites (*L. deliense*, *L. indicum*, *S. ligula*, *L.keukenschrijveri*, *L. rajesthanensis*, *T.hypodermata*, *Microtrombicula* spp., *and N. microti*) infestation in domestic and peri-domestic areas [6].

### 2.2. Sample Collection

The blood sample collected from the patient clinically suspected of *St* infection who was seeking medical assistance from the Government Medical College & Hospital at Thanjavur has been utilized for the detection of genotype. After serological diagnosis by commercial IgM-ELISA kits (Scrub Typhus Detect™ IgM ELISA, IndBios International, Inc., Seattle, WA, USA), the positive samples (*n* = 59) were subjected to the genome analysis study. A molecular genotypic assay was carried out at the ICMR-Vector Control Research Centre, Field Unit. The necessary ethical clearance was obtained from the Institutional Human Ethical Committee (TMC/IHEC: 1118/2023; 27 July 2023).

### 2.3. Detection of O. tsutsugamushi by Nested PCR

The Genomic DNA was extracted from 59 serum samples using the commercially available QIAamp DNA Blood Mini Kit (Cat No. 51104, Qiagen, Hilden, Germany). The DNA extracted was subjected to a nested polymerase chain reaction (nPCR) assay targeting the *Ot* GroEL gene as per the protocol described by Weihong Li et al. [32]. Two µL of DNA was used as a template for the first round of PCR with the primers Gro-1, 5′-AAGAAGGACGTGATAAC-3′ and Gro-2, 5′-ACTTCACGTAGCACC-3′. Subsequently, one microliter of the first-round PCR product was further used as the template for nested PCR with the primers TF1, 5′-ATATATCACAGTACTTTGCAAC-3′ and TR2, and 5′-GTTCCTAACTTAGATGTATCAT-3′ to amplify a product of 365 bp of the GroEL gene fragment in a final volume of 25 µL for both PCR reactions. In the post-PCR processing, the amplified DNA fragment was resolved in a 2% agarose gel by loading 5 µL of the PCR product with the 100 bp DNA ladder and a negative (water) control loaded alongside. The remaining amplified gene fragment was further subjected to Sanger DNA sequencing in both forward and reverse directions.

### 2.4. PCR Amplification of 56 kDa VDI-III

All fourteen samples amplified with the primer specific for the GroEl gene were subjected to nested PCR amplification of the 56 kDa gene, with the following outer and inner primers: JG-OtF584 (5′-CAA TGT CTG CGT TGT CGT TGC) and RTS9 (5′-ACAGAT GCA CTA TTA GGC AA); F (5′-AGC GCTAGG TTT ATT AGC AT) and RTS8 (5′-AGG ATT AGA GTG TGG TCCTT), respectively, as described by Toon Ruang-areerate et al. [33]. Then, 5 µL of the PCR product was loaded in 1% agarose gel, and after the confirmation of bands, the remaining PCR-amplified products were purified and custom DNA-sequenced.

### 2.5. DNA Sequencing and Phylogenetic Analysis

The DNA sequencing chromatograms were manually examined and edited by comparing both forward and reverse DNA strands. The final DNA sequences thus obtained were subjected to the nucleotide BLAST sequence analysis tool in the NCBI database to determine the sequence similarity. Fourteen partial GroEL and twelve 56 kDa DNA sequences generated in this study were deposited in the NCBI GenBank database, and accession numbers were obtained (GenBank Acc. nos. GroEL: OR887439-OR887452; 56 kDa: PQ381691-PQ381702). The nearest-neighbor sequences exhibited low e-value and higher nucleotide identity in BLAST analysis against the respective GroEL and 56 kDa study sequences. In addition, the *Ot* genotype-specific reference sequences belonging to Karp, Kato, Gilliam-types, and TA716 strains were also included to calculate the nucleotide diversity (d) and check the phylogenetic relationship among them. Phylogenetic analysis was conducted using the Molecular Evolutionary Genetics Analysis [MEGA 11] software [34]. The mean nucleotide diversity of the entire GroEL study sequences, between and within the genotypes, was calculated after grouping the DNA sequences based on the phylogenetic clades.

Similarity profile and bootscan analysis for checking the similarity of the study sequences with neighboring reference sequences and to detect the event of recombination among the strain was carried out by the Simplot software 3.5.1 [35]. Only three prototype Karp, Kato, and TA716 sequences were included to avoid noise during the Simplot analysis. A neighbor-joining similarity, as well as a recombinant distance plot, was drawn with a nucleotide substitution model of Kimura 2 parameter (K2P) and a window and step size of 200 and 20, respectively. The consensus output was obtained by computing 1000 bootstrap replicates, and the similarity and recombination were detected among the study sequences.

## 3. Results

### 3.1. GroEL Gene Sequence Analysis

A total of 59 samples were processed by nested PCR; the GroEL gene-specific product (361 bp) was amplified in 14 (23.7%) samples. The analysis of the fourteen DNA sequences showed three distinct phylogenetic clades, viz new *Ot*-TJTN, Karp-like, and Kuroki-Gilliam types in the neighbor-joining (NJ) dendrogram (Figure 1a).

Further, to check the consistency of the phylogenetic inference obtained in the NJ tree, the Maximum Likelihood tree (ML) and Maximum Parsimony tree (MP) were constructed with the same DNA sequences, which showed consistent genotypic clades as obtained in the NJ tree. Among the 14 *O. tsutsugamushi* sequences generated in this study, 9 (64%) sequences formed a distinct new genotypic cluster related to the TA716-like Thai strain, and 4 (28.5%) and 1 (7%) sequence formed Karp-like and Kuroki-Gilliam types, respectively (Figure 1a and Figure 2).

The percentage of nucleotide identity for the nine sequences was 100% with the DNA sequences submitted from South India, GenBank Acc. nos. ON156002 and CP166954, followed by 97.71% (EF551288) from Thailand and 97.43% (CP044031; LS398552) from China and the UK, respectively. Two out of four sequences in the Karp-like clade were found to be identical, and the other two (OR887441 and OR887446) differed by two and one nucleotide substitution, respectively (Table 1).

### 3.2. TSA 56 kDa Gene VDI-III Sequence Analysis

All 14 GroEL PCR-positive samples were subjected to amplification of TSA 56 kDa variable domain-I-III. During the post-PCR processing, two different-sized fragment patterns of the PCR-amplified 56 kDa gene were recorded. Out of 14, a total of 12 TSA 56 kDa DNA sequences were generated and subjected to multiple sequence analysis in MEGA.11 software. DNA sequences having the query coverage of 99–100% with 98.3% to 99.9% nucleotide identity were retrieved from the GenBank database, and a conventional as well as a radiation distance phylogenetic tree was constructed. The resulting dendrogram with the 56 kDa sequences retrieved from GenBank showed a distinct cluster of *Ot*-TJTN and another Karp-like genotype (Figure 1b and Figure 3). The one Gilliam-type *Ot* sequence obtained in GroEL PCR did not amplify in the 56 kDa PCR assay. Among the twelve 56 kDa sequences, eight sequences of the new *Ot*-TJTN formed as a separate monophyletic cluster with the *Ot* 56 kDa gene sequences from Vellore, South India, and the sequences from Taiwan (CP142420, MW495758, MW495734) formed as a sub-clade in the Kato genogroup (Figure 1b and Figure 3).

The TA716 and Kato genotypes are a common ancestral group to the newly emerged lineage. The other four Karp-like sequences (PQ381691, PQ381693, PQ381697, and PQ381700) formed as a monophyletic group along with the reference strain and sequences from India, Taiwan, China, and Cambodia. Individual DNA sequences were analyzed in BLAST, and information such as the GenBank accession numbers, length of query coverage, nucleotide identity, source of isolation of *Ot*, and the country from where the sequence was deposited were collected and presented in Table 2. 

### 3.3. TSA 56 kDa Gene VDI-III Similarity Plot Analysis

Only the eight *O. tsutsugamushi* 56 kDa gene sequences identified as a new cluster in this study were subjected to Simplot analysis with the 56 kDa strain-specific reference sequences. The similarity plot values of −2.18, −3.06, and −5.06 indicated that the strain TA716-Thai isolate is related to 56 kDa *Ot* sequences generated in this study, followed by being distantly related to the Karp and Kato genotypes, respectively. The result of the bootscan plot demonstrated the event of possible recombination between the TA716-like strain and Kato reference sequences at the nucleotide positions 373 and 655 of the 56 kDa variable domain I-III (Appendix A).

## 4. Discussion

Scrub typhus (St), one of the neglected tropical diseases, has recently been reported to be escalating in case incidences in endemic areas. Different molecular markers are being used for the identification of the *Ot* genotypes [15]. However, to our limited knowledge, this is the first report on exploiting the two markers of GroEL and 56 kDa genes to screen, analyze, and determine the phylogenetic relationship of the *Ot* recovered from human St cases from the southeastern part of India. In our earlier study, the GroEL gene sequence-based phylogenetic analysis detected the three distinct genotypes of *Ot* in the Theni district, which is an adjacent district in Tamil Nadu [30]. Interestingly, 9 out of 14 sequences generated in this study were found to have a higher percentage of nucleotide identity with the *Ot* DNA sequences available in the NCBI GenBank database. Among the identical *Ot* DNA sequences retrieved, the following GroEL sequences with GB Acc. Nos. ON156002, CP166954, CP166956, and PP754991 generated from human and rodent (unpublished) samples, respectively, from Tamil Nadu, South India were found to be 100% identical, followed by EF551288 (TA716 Thailand) and LS398552 (UT76 China) at 97.71% and CP044031 (Wji/2014 China) at 97.41%. Six out of nine sequences in the new *Ot*-TJTN strain generated were found to be 100% identical, which probably indicates their origin from a common ancestor (Figure 1a and Figure 2). However, this needs further confirmation.

In the other GroEL sequence cluster, 4 out of 14 sequences generated were found to have 99.14% to 99.71% of nucleotide identities with Karp-like sequences from the United Kingdom, India, Thailand, Japan, and the USA. Among the closely related sequences retrieved from the genome database, the flowing sequences, viz ON156000 VCRC/India, EF551309 Thailand, and JX188400 Japan, were found to have the highest nucleotide identity of 99.71% (Table 1). In the other clade, a single sequence out of 14 sequences was found to have a nucleotide identity of 96.29% to 100% with the Kuroki-Gilliam-Boryong-Kawasaki DNA sequences retrieved from the genome database (Figure 1a). Among them, the Kuroki-Gilliam (ON156004) strain detected from human cases in the Theni district, South India, belonging to the earlier study was 100% identical [30] (Table 1). This result further indicates the introduction of a strain derived from common ancestry that is circulating in the southwestern district of Theni, which is situated about 250 km apart from the present study area in the southeastern district of Thanjavur.

The intra-strain mean diversity (d) was found to be higher (0.004) in the Karp-related group when compared with the new *Ot*-TJTN (0.001) and Kuroki (0.000) groups. The intra-mean diversity indicates that the *Ot*-TJTN strain is less diverse than the Karp-related strain. A single Kuroki-Gilliam-type sequence from this study (OR887445) was grouped with a Kawasaki-related sequence (ON156004 VCRC/India); upon analysis, there was no nucleotide diversity detected within this group. Further, the inter-strain mean diversity was calculated after including reference sequences [29] and closely related sequences retrieved from the genome database in the appropriate group. The results showed a mean diversity of *d* = 0.031 between TJTN-Karp-like & *Ot*-TJTN-Kato-related strains, and the highest diversity of *d* = 0.052 was calculated between the TJTN and Kuroki-Gilliam types. These *d* values indicate the independent circulation of the TJTN strain in this region.

In this study, analyses of hypervariable domain I-III (VDΙ-ΙΙΙ) demonstrated the possible circulation of a new strain. The length of the DNA fragments was 650 bp and 692 bp for the *Ot*-TJTN and Karp-like sequences, respectively. The deletion of 42 nucleotides in the new genotypic clade resulted in amino acid variation when compared with the related reference strains included in the analysis and Karp-like sequences detected in this study (Figure 4).

TSA 56 kDa amino acid similarity analysis in BLASTP showed 98% to 100% with the *Ot* strain sequences submitted from north Tamil Nadu (XDE91401–XDE91404), followed by 86% identity with the TSA sequences from Japan (AA072147). The neighbor-joining conventional and radiation dendrograms for the 56 kDa show two genotypic clusters (Figure 1b and Figure 3). The Karp-like sequences exhibit a higher percentage of query coverage and similarity from different countries in Southeast Asia. On the other hand, the new genotype showed a 100% similarity to the isolates from South India and 98.5% and 92.6% identity with *Ot* isolated from humans and chiggers from Taiwan, respectively. The above analysis confirms the generation of different clones due to genetic recombination in the chigger mites and the circulation of region-specific strains [7,33].

Simplot analysis for testing the nucleotide similarity and bootscan analysis for detection of a recombination event between the study sequences (designated as query sequences) and prototype reference sequences showed a partial overlapping curve in the TA716 and Kato strain at the VD-I. The bootscan analysis showed a possible recombination event between TA716 and Kato at nucleotide positions 373 and 655 of the TSA 56 kDa gene. The Simplot test supports the result of phylogenetic analysis in concordance (Appendix A), and the detected new cluster is genetically related to the Kato and TA716 lineage but divergent to the extent of the new strain.

The results of the earlier studies have demonstrated the dominant occurrence of *Ot* belonging to the Kato-like strain in many parts of India [4,25]. However, the evidence of the present study shows the distribution of Karp-like and the novel *Ot*-TJTN genotypes. In India, the results of 56 kDa TSA gene sequence-based genotype studies have been analyzed, and it was found that the TA678 strains had close genetic similarity to the Koto and TA716 genotypes [4,24]. Interestingly, *O. tsutsugamushi* isolated from rodents (P754990, P754991 unpublished) and human cases (ON156002, ON156003) from Theni districts matched 100% nucleotide identity with the TJTN GroEL sequences in this study [30]. Notably, the nine GroEL and the eight 56 kDa TSA sequences generated from this study formed a separate cluster, and they did not form a cluster with either the Koto genotypes or Koto reference sequences available in the GenBank database. The higher GroEL nucleotide percentage (97.71%) identity was obtained with EPW1038 of the TA716 (EF551288) Thailand reference strain next to the ON156002 (100%) generated from this geographic region. Similarly, the 56 kDa sequences also did not cluster with the sequences belonging to different genotypes, except the 56 kDa sequences generated from this geographic region, followed by Taiwan isolates with 92.6% to 92.9% (Table 2, CP142420, MW495758, and MW495734). The result of these analyses indicates that all nine GroEL and the eight 56 kDa *Ot*-TJTN sequences are undescribed but closely related to the TA716 Thai isolate (Figure 2 and Figure 3). Further, the result implies that the *Ot*-TJTN strain is circulating in humans and rodents and has been recorded for the first time in this area. It is worth understanding their possible regional-specific distribution dynamics, if any.

Worldwide, in recent days, molecular genotype studies have revealed the distribution of different genotypes. The close identity was drawn mainly in comparison with the major prototypes of Karp, Kato, Gilliam, Boryong, Ikeda, Kawasaki, Kuroki, Saitama, and Shimokoshi available in the genome database. Interestingly, *Ot* genotypes identified in this study show close identity with strains in the SEA region (Figure 1), which confirms the circulation of ancestral haplotypes throughout the SEA continent [33]. Therefore, these strains may be considered to be specific to this geographical region because the antigenic heterogeneity is restricted to the geographic epicenter of St endemicity [36].

## 5. Conclusions

The results of this preliminary study demonstrate a rich genetic diversity in *O. tsutsugamushi* in and around Thanjavur district, Tamil Nadu, India. The phylogenetic analysis based on the partial genome sequence of the GroEL and 56 kDa genes shows the circulation of a novel strain, *Ot*-TJTN, along with Karp-like and Kuroki-Gilliam-type strains. Further, the result shed light on conducting a genome-based study on the scrub typhus pathogen, considering the increased scrub typhus cases [11,12], neurological complications, and emergence of new species [17,18,37]. Also, the outcome of this study warrants further studies on understanding the mechanisms of the ongoing transmission dynamics of *Ot* strains in vector mites, rodent hosts, and humans for implementing appropriate control measures, particularly one health approach in the future.

## Figures and Tables

**Figure 1 microorganisms-13-00333-f001:**
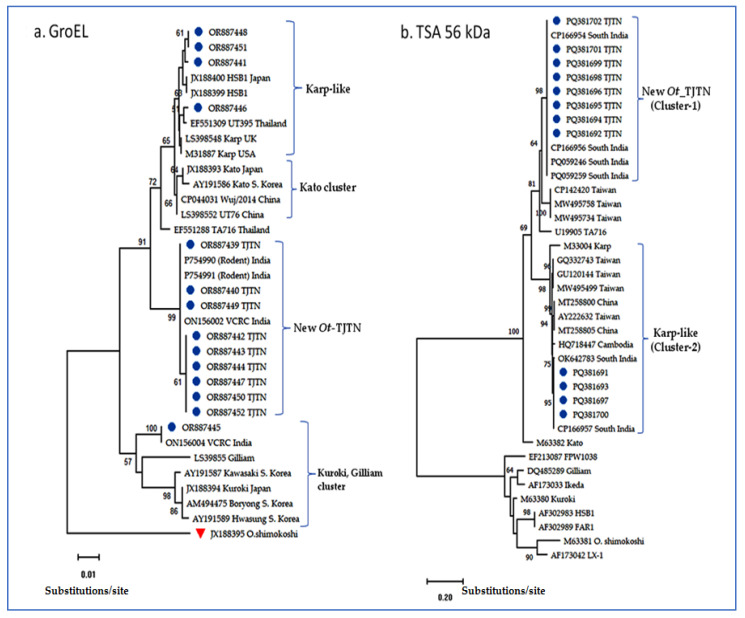
Neighbor-joining (NJ) tree of the GroEL (**a**) and 56 kDa TSA gene (**b**) with 1000 bootstrap replicates, K2P substitution model. The 56 kDa gene (**b**) with reference sequences showing two clusters, a novel TJTN and Karp-like. Sequences obtained from this study were labeled with a blue circle; the out-group sequence is marked with a red triangle for GroEL (**a**).

**Figure 2 microorganisms-13-00333-f002:**
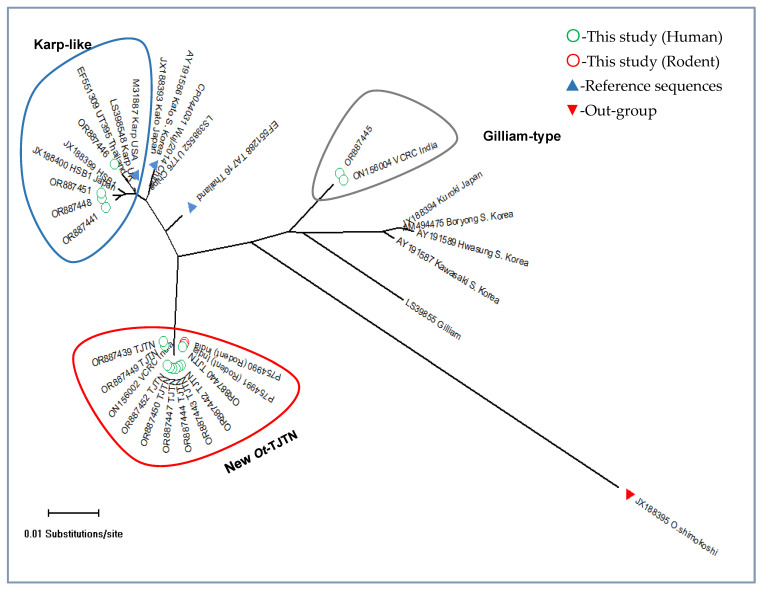
A radiation distance phylogenetic tree of GroEL gene sequences showing the three different clusters, and the designated new *Ot*-TJTN is clustered (red) separately, deviating from the Karp, Kato, and TA716 (blue) and Gilliam clade (grey).

**Figure 3 microorganisms-13-00333-f003:**
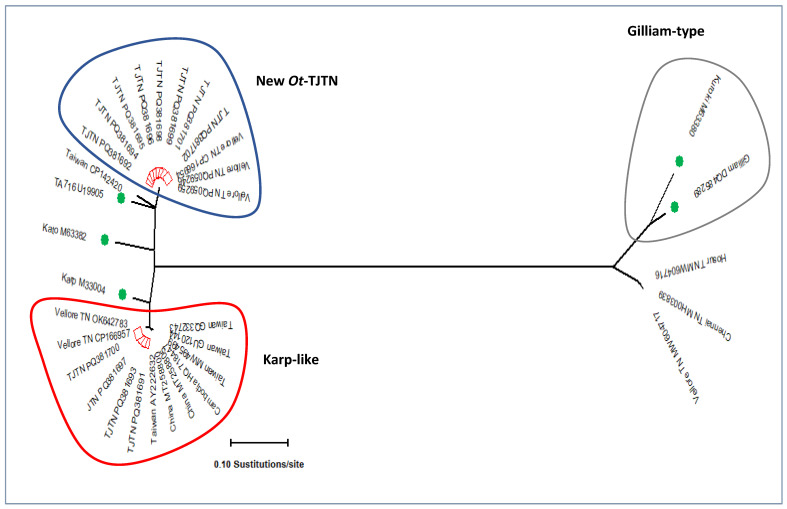
A radiation distance phylogenetic tree of the 56 kDa TSA sequences showing the two different clusters of designated new *Ot*-TJTN and Karp-like genotypes. Kato and TA716 were ancestral lineage to the novel *Ot*-TJTN genotype (blue) and Karp-like sequences (red), and the Gilliam and Kuroki genotypes formed a separate cluster. (Square in red: sequences from this study and circle in green: reference sequences).

**Figure 4 microorganisms-13-00333-f004:**
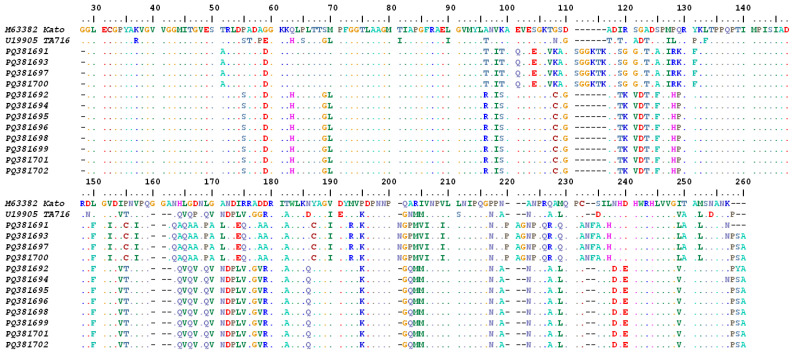
Multiple sequence alignment of the Karp-like and *Ot*-TJTN 56 kDa TSA amino acid sequences generated in this study with the closely related reference Kato and TA716 genotypes. The alignment shows the non-synonymous change of amino acids in 56 kDa protein VD-ΙI-III due to the nucleotide substitutions.

**Table 1 microorganisms-13-00333-t001:** Details of samples, patients, GenBank accession numbers of GroEL gene sequence, and percentage identity to query sequences.

Sl No.	DNA Code	GenBank Acc. Nos. (Present Study)	Isolates/Strains	Closest GenBank Acc. Nos.	Nucleotide Identity (%)	Country
*O. tsutsugamushi* new genotype (*Ot*-TJTN, 9/14)						
1	TJ09	OR887439	VCRC/THRLU2	PP754991	100	India
2	TJ11	OR887440	JJOtsu6	CP166956	100	India
3	TJ13	OR887442	VCRC/GH-8252	ON156002	100	India
4	TJ24	OR887443	JJOtsu1	CP166954	100	India
5	TJ25	OR887444	TA716	EF1551288	97.71	Thailand
6	TJ35	OR887447	Wuj/2014	CP044031	97.43	China
7	TJ39	OR887449	UT76	LS398552	97.43	China
8	TJ42	OR887450	VCRC/THRLU2	PP754991	99.71	India
9	TJ147	OR887452	VCRC/GH-8252	ON156002	100	India
Karp-like (04/14)						
10	TJ12	OR887441	HSB1	JX188400	99.43	Japan
			2049	EF551291	99.43	Thailand
11	TJ33	OR887446	UT395	EF551309	99.71	Thailand
12	TJ38	OR887448	VCRC/GH-4505	ON156000	99.71	India
			Karp	LS398548	99.14	UK
Kuroki-Gilliam cluster (01/14)						
14	TJ26	OR887445	VCRC/GH-6202	ON156004	100	India
			Kuroki	JX188964	96.86	Japan
			Boryong	AM494475	96.86	S. Korea
			Kawasaki	AY191587	96.86	S. Korea
			Hwasung	AY191589	96.57	S. Korea
			Gilliam	LS398551	96.29	UK

**Table 2 microorganisms-13-00333-t002:** Details of samples, GenBank accession numbers of 56 kDa gene sequence, and percentage identity to query sequences.

Sl. No.	DNA Code	GenBank No. (This Study)	Query Coverage (%)	Identity (%)	Source	Country	Closest GenBank Acc. No.
Group 1: Karp and TA716 like							
1	TS1	PQ381691	100	99.9	Human	South India	CP166957
			99	98.5	Human	Taiwan	GQ332743
			99	98.5	Chigger	Taiwan	GU120144
2	TS3	PQ381693	100	99.9	Human	South India	CP166957
			99	98.6	Human	Taiwan	MW495499
3	TS7	PQ381697	100	99.9	Human	South India	CP166957
			99	99.7	Human	South India	OK642783
4	TS10	PQ381700	100	99.9	Human	South India	CP166957
			99	98.4	Human	Cambodia	HQ718447
			99	98.5	Human	China	MT258805
			99	98.3	Human	Taiwan	AY222632
Group 2: New Genotype (*Ot*-TJTN)							
5	TS2	PQ381692	98	100	Human	South India	CP166956
			99	92.6	Human	Taiwan	CP142420
6	TS4	PQ381694	99	100	Human	South India	PQ059254
			99	99.8	Human	South India	PQ059257
			99	92.8	Human	Taiwan	MW495758
7	TS5	PQ381695	99	100	Human	South India	PQ059254
			99	92.6	Human	Taiwan	CP142420
8	TS6	PQ381696	99	100	Human	South India	CP166956
			99	92.9	Human	Taiwan	MW495734
			99	92.8	Human	Taiwan	MW495758
9	TS8	PQ381698	98	100	Chigger	South India	CP166954
			99	92.6	Human	Taiwan	CP142420
10	TS9	PQ381699	98	100	Human	South India	CP166956
			99	92.9	Human	Taiwan	MW495734
11	TS11	PQ381701	99	100	Human	South India	PQ059254
			99	98.8	Human	South India	PQ059259
			99	92.9	Human	Taiwan	MW495734
12	TS12	PQ381702	98	100	Chigger	South India	CP166954
			91	100	Human	South India	MH003839
			99	100	Human	South India	PQ059259
			99	92.9	Human	Taiwan	CP142420

## Data Availability

The original contributions presented in the study are included in the article/Appendix A, further inquiries can be directed to the corresponding authors.

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
