# Peer review of "A Novel Genotype of Orientia tsutsugamushi in Human Cases of Scrub Typhus from Southeastern India"

_microorganisms, 2025, doi:10.3390/microorganisms13020333_

Round 1

Reviewer 1 Report

Comments and Suggestions for Authors

Dear Authors, 

The paper presents a promising study on a new strain of O. tsutsugamushi. However, some details need to be corrected. Several paragraphs in the introduction are missing references. In addition, there are some English errors that need to be revised. Furthermore, the first map needs to be revised, as it does not mean anything. Otherwise, the paper was well structured, with a well-explained methodology, followed by correct results and discussion.

Comments on the Quality of English Language

A native speaker need to read all the text.

Author Response

Response to Reviewer 1 Comments

 Open Review

Comments and Suggestions for Authors

Dear Authors,

Comments 1

The paper presents a promising study on a new strain of O. tsutsugamushi. However, some details need to be corrected. Several paragraphs in the introduction are missing references.

Response 1: Revision marked in the .pdf file was carried-out and references were incorporated in appropriate points

Comments 2

In addition, there are some English errors that need to be revised.

Response 2: Agreed. Revised as per the suggestion

Comments 3

Furthermore, the first map needs to be revised, as it does not mean anything.

Response 3: Accepted. Map of study area revised with new map

Reviewer 2 Report

Comments and Suggestions for Authors

In the reviewed MS, a group of scientists from India provided results of their study on genetic variability of the pathogenic bacteria Orientia tsutsugamushi, the causative agent of Scrub typhus vectored by mites. They investigated 59 blood samples positive for O. tsutsugamushi and performed PCR screening using two gene markers: GroEL and f 56 kD TSA protein. They performed a series of analyses, revealed a new local genotype of O. tsutsugamushi and speculated on its relations with other genotypes of O. tsutsugamushi (sequences from GB). The MS is poorly written and structured. The MS is very long, it has very simple content and could be three times shorter. The analyses performed, described and interpreted in suboptimal way. The authors used old software for detecting recombination. They did not describe the applied methods in detail and did not explain how the datasets for phylogenetic analyses were constructed and how the sequences were aligned. Some Figures are redundant. All figures and captions need revision. The molecular phylogenetic analyses should be redone with inclusion of larger number of sequences from GB. The Discussion is too long. It includes lots of unnecessary information and because of this is poorly focused on the goals of the MS. The MS needs moderate linguistic corrections. The MS needs serious/fundamental revision.

Title: it is very long. No circulation was actually shown in this study, therefore n oneed to mention it in the title. Possible title: “A novel genotype of Orientia tsutsugamushi in Human Cases of Scrub Typhus from Southeastern India”

Abstract: variable domains (VDI-VDIII) --- what is this? not clear

The last 2 sentences of the Abstract need rewording

55. Leptotrombidium sp, Schoengastiella sp, and Ornithonyssus sp. – instead of “sp.” use “spp.”

57 Southeast Asian countries – please give a list in brackets and a reference

68 (Orientia tsutsugamush - italic

73 key strains – please explain biefly why they are considered “key” and based on which markers they were designated

74 prototype strains – please explain this for unprepared readers

86. O. – start a sentence with full generic name

88-90 revise this sentence

90. Please explain better what do you call “variant”

91-99 this is too long text for a very simple idea. Please, consider revising

103. please give some geographic data for this hospital

104-105 it is not clear how “genomic studies” and “circulating” are connected. Please, revise this sentence

Intro: last two paragraphs need serious revision. Please indicate the main goals of the study. Please unambiguously explain what previous authors achieved and how your new research complements their data.

Figure1. Please provide a new version of this figure. Please use scientific style when preparing a figure. Please explain all red ellipses (why they are ellipses, not circles?). Please, revise the caption. What are the “related strain”? What do you mean? related to what?

119. rice bawl – redundant

120 rodent – please, give a brief list of rodent and mite genera, registered in this area and important for your research

145,146 AAGAAGGA/CGTGATAAC, ACTTCA/CGTAGCACC – not clear what the oblique slash line means

183 Phylogenetic analysis  - please, describe the analyses: method, model, adjustments… This is important since your analysis should be repeatable if anybody would like to reanalyze your dataset. Please, explain how you composed the datasets which was used for mol.phyl. analyses and the out-group choice.

191. Simplot is a very old Software. Please, explain the choice of this method. Why you did not use more modern program, e.g. RDP?

191. Only three prototype Karp, Kato, and TA716 – this is not clear. If you want to detect the recombination event you should include as much different genotypes as possible to avoid possible absence of “parents”. Please, explain this and perform a new analysis

Figure 2. This tree needs revision. Please, include al main genotypes that are available in GB and show the clustering of your new sequences with other GB sequences. The text (lines 201-213) and the tree on Fig2 do not coincide. please redo the analysis and redescribe it. What are the green circles in the tree?

220-231 Please rewrite this paragraph. No need to repeat the data given in the Table 1. Please give only the main conclusions/overview.

Table 2. Please, revise this Table. What is bold and underlined? Are these sequences present in the mol.phyl trees? Why some lines in the Table are empty?

261-262: This table needs revision as well as the entire mol.phyl. analysis. Check lines in different columns, they should be match to each other

264: Abbreviations – redundant

Fig.3 This figure is technical, it could be given in a Supplement

285-308. This information repeats Intro. Please consider transferring it to Introduction.

295 vs 309-311 – contradiction

Fig.4 Why Fig.4 is given in Discussion? Please, explain how was it obtained? What different colors and the arrow mean?

Discussion. This section needs complete rewriting. It is a mess.

Conclusions. Current Conclusions is far from the exact results obtained in the study. Please, provide distinct conclusions supported by your results.

Author Response

    In the reviewed MS, a group of scientists from India provided results of their study on genetic variability of the pathogenic bacteria Orientia tsutsugamushi, the causative agent of Scrub typhus vectored by mites. They investigated 59 blood samples positive for O. tsutsugamushi and performed PCR screening using two gene markers: GroEL and f 56 kD TSA protein. They performed a series of analyses, revealed a new local genotype of O. tsutsugamushi and speculated on its relations with other genotypes of O. tsutsugamushi (sequences from GB). The MS is poorly written and structured. The MS is very long, it has very simple content and could be three times shorter. The analyses performed, described and interpreted in suboptimal way. The authors used old software for detecting recombination. They did not describe the applied methods in detail and did not explain how the datasets for phylogenetic analyses were constructed and how the sequences were aligned. Some Figures are redundant. All figures and captions need revision. The molecular phylogenetic analyses should be redone with inclusion of larger number of sequences from GB. The Discussion is too long. It includes lots of unnecessary information and because of this is poorly focused on the goals of the MS. The MS needs moderate linguistic corrections. The MS needs serious/fundamental revision.

Comments 1: Title: it is very long. No circulation was actually shown in this study, therefore no need to mention it in the title. Possible title: “A novel genotype of Orientia tsutsugamushi in Human Cases of Scrub Typhus from Southeastern India

Response 1: We agree with the reviewer suggestion.

Comments 2: Abstract: variable domains (VDI-VDIII) --- what is this? not clear

Response 2: Revised as TSA 56 kDA gene variable domains (VDI-III)

Comments 3

The last 2 sentences of the Abstract need rewording

Response 3: Revised as suggested

Comments 4

  1. Leptotrombidium sp, Schoengastiella sp, and Ornithonyssus sp. – instead of “sp.” use “spp.”

Response 4: Revised as per suggestion.

Comments 5

57 Southeast Asian countries – please give a list in brackets and a reference

Response 5: Names of the SEA countries incorporated with reference

Comments 6

Line 68 (Orientia tsutsugamushi) – italic

Response 6:

Revised as per suggestion

Comments 7

Line 73 key strains – please explain biefly why they are considered “key” and based on which markers they were designated

Response 7:

Revised as per suggestion and incorporated in the text.

Comments 8

Line 74 prototype strains – please explain this for unprepared readers

Response 8:

Revised as per suggestion and incorporated in the text

Comments 9

  • – start a sentence with full generic name

Response 9:

Revised as per suggestion

Comments 10

Line  88-90 revise this sentence

Response 10 :

Revised as per suggestion

Comments 11

Line 90 Please explain better what do you call “variant”

Response 11:

“Variant” term was deleted to avoid confusion between the genotype and variant.

Comments 12

91-99 this is too long text for a very simple idea. Please, consider revising

Response 12

 Revised as per suggestion  

Comments 13

  1. please give some geographic data for this hospital

Response 13

Detailed geographical information provided in the material and methods (sub heading , Study area)

Comments 14

104-105 it is not clear how “genomic studies” and “circulating” are connected. Please, revise this sentence

Response 14:

We agree with the reviewer. Appropriate term has been incorporated to replace “genomic studies”

Comments 15

Intro: last two paragraphs need serious revision. Please indicate the main goals of the study.

Response 15 :

Reviewers’ suggestion has been incorporated and revised.

Comments 16

Please unambiguously explain what previous authors achieved and how your new research complements their data.

Response 16

The result of our previous study was explained unambiguously

Comments 17

Figure1. Please provide a new version of this figure. Please use scientific style when preparing a figure.

Response 17

Revised as per reviewers suggestion

Comments 18

Please explain all red ellipses (why they are ellipses, not circles?). Please, revise the caption. What are the “related strain”? What do you mean? related to what?

Response 18

Figure 1 was replaced with a revised

Comments 19

  1. rice bawl – redundant

Response 19:

The river Cauvery Delta is often referred to as the "Rice Bowl" of Tamil Nadu due to its extensive paddy cultivation

Comments 20

  1. 120 rodent – please, give a brief list of rodent and mite genera, registered in this area and important for your research

Response 20

As suggested, both rodent and infesting mite species names were incorporated

Comments 21

145,146 AAGAAGGA/CGTGATAAC, ACTTCA/CGTAGCACC – not clear what the oblique slash line means

Response 21

“/” was deleted

Comments 22

183 Phylogenetic analysis - please, describe the analyses: method, model, adjustments… This is

important since your analysis should be repeatable if anybody would like to reanalyze your dataset.

Response 22

 Details of phylogenetic analysis as suggested was incorporated

Comments 23

Please, explain how you composed the datasets which was used for mol.phyl. analyses and the outgroup choice.

Response 23

The DNA sequence chromatogram was manually inspected and the reverse sequences chromatogram was copied into reverse complimentary form. Once, the full length sequence was finalized, the sequence was saved with sample ID. The final sequence was individually blast analyzed and the identical sequences to the study sequences were copied to a dataset. A final dataset was created and subjected for multiple sequences alignment of only study sequences as well as closely related sequences retrieved from GenBank database. After incorporating the reference and out-group sequences, the multiple sequences alignment was saved as .MEGA format. The same file was used for the analysis with defining the common window for sequences to be analyzed. For calculating nucleotide diversity, sequences were grouped according to the genotype. The neighbour joining tree was constructed with K2P model, 1000 bootstrap replicate. Labelling with different colour was done after construction of phylogenetic tree. Based the final tree, the result and interpretation was completed.    

Comments 24

  1. Simplot is a very old Software. Please, explain the choice of this method. Why you did not use more modern program, e.g. RDP?

Response 24

We agree with the suggestion of the reviewer. The result of phylogenetic analysis was supported by simplot analysis. Hence, we did not search for additional one. As suggested, the Simplot images may be accommodated in the supplementary file.

Comments 25

  1. Only three prototype Karp, Kato, and TA716 – this is not clear. If you want to detect the

recombination event you should include as much different genotypes as possible to avoid possible absence of “parents”. Please, explain this and perform a new analysis

Response 25

Our sequences highest identity was obtained within the three genotypes hence, we did not include.

Comments 26

Figure 2. This tree needs revision. Please, include al main genotypes that are available in GB and show the clustering of your new sequences with other GB sequences. The text (lines 201-213) and the tree on Fig2 do not coincide. please redo the analysis and redescribe it. What are the green circles in the tree?

Response 26

We agree with the reviewer, All the suggestion were attended and the phylogenetic tree (GroEl and 56kDa) were revised

Comments 27

220-231 Please rewrite this paragraph. No need to repeat the data given in the Table 1. Please give only the main conclusions/overview.

Response 27

Suggested revision was made

Comments 28

Table 2. Please, revise this Table. What is bold and underlined? Are these sequences present in the mol.phyl trees? Why some lines in the Table are empty?

Response 28

The bold and underline were removed.

No. of sequences matching with the study sequences and retrieved from GenBank data base are more. Hence, the group of study sequences was aligned in canter of the column.

Comments 29

261-262: This table needs revision as well as the entire mol.phyl. analysis. Check lines in different columns, they should be match to each other

Response 29

Suggestion of the reviewer is considered. However, to avoid repetition of GenBank sequence against the individual sequence a common matching sequence were provided in the next column.

Comments 30

264: Abbreviations – redundant

Response 30

Revised with clear writing

Comments 31

Fig.3 This figure is technical, it could be given in a Supplement

Response 31

We agree with the reviewer suggestion

Comments 32

285-308. This information repeats Intro. Please consider transferring it to Introduction.

Response 31

We agree with the reviewer suggestion. Deleted.

Comments 32

295 vs 309-311 – contradiction

Response 32

In the earlier study, only GroEL gene was used with limited number of sample. But, to avoid ambiguity on determining the genotype we used an additional marker (TSA56 kDa gene)

Comments 33

Fig.4 Why Fig.4 is given in Discussion? Please, explain how was it obtained? What different colors and the arrow mean?

Response 33

 Figure 4 was moved from discussion to result part and the radiation tree was drawn by using MEGA software. The red batch is new genotype and the arrow indicates the closely related genotype to the Ot-TJTN. Same information was incorporated in the figure caption.

Comments 34

Discussion. This section needs complete rewriting. It is a mess.

Response 34

Suggestion was taken care and revised

Comments 35

Conclusions. Current Conclusions is far from the exact results obtained in the study. Please, provide distinct conclusions supported by your results.

Response 35

Suggestion was taken care and revised

Reviewer 3 Report

Comments and Suggestions for Authors

Abstract :

Refine the abstract to provide a clearer summary of the main findings and their implications. Highlight the key results and their relevance more succinctly.

Introduction :

The introduction could benefit from more detailed comparisons to existing studies in other regions or countries, enhancing the global context of the research.

Result :

Elaborate on the clinical relevance of the new Ot-TJTN genotype. Discuss how these findings might influence diagnostics, treatments, or public health strategies.

Provide more insight into the significance of the observed nucleotide diversity within and between groups.

Discussion :

Compare the findings more extensively with related studies, not just regionally but also internationally.

Highlight the potential implications of the novel Ot-TJTN genotype on vaccine development or diagnostic panels.

Address any limitations in the study, such as sample size or geographic focus, and suggest future research directions.

Concluding :

Strengthen the conclusion by emphasizing how the findings contribute to current knowledge and what specific steps should be taken next.

Author Response

Thank very much fro the valuable comments and suggestion 

  1. Point-by-point response to Comments and Suggestions for Authors

Comments 1:

Abstract : Refine the abstract to provide a clearer summary of the main findings and their implications. Highlight the key results and their relevance more succinctly

Response 1: Agreed with the suggestion of the Reviwer.1 and the ABSTRACT has been revised incorporating the suggested points

Comments 2: The introduction could benefit from more detailed comparisons to existing studies in other regions or countries, enhancing the global context of the research.

Response 2: Agreed with the reviewer and the suggestions has been incorporated in the Text.

Comments 3 Result : Elaborate on the clinical relevance of the new Ot-TJTN genotype. Discuss how these findings might influence diagnostics, treatments, or public health strategies. Provide more insight into the significance of the observed nucleotide diversity within and between groups.

Response 3. As this is the first time we are proposing the New strain namely; Ot-TJTN circulation behind clinical human cases in this region, in future its other clinical relevance such as its virulence, severity among human cases will be meticulously studied. Hence the suggested point is not elaborated in the text.

The other suggestion of discussing the findings might influence diagnostics, treatments, or public health strategies and the significance of the observed nucleotide diversity has been discussed and incorporated in the text.

Comments 4: Discussion: Compare the findings more extensively with related studies, not just regionally but also internationally. Highlight the potential implications of the novel Ot-TJTN genotype on vaccine development or diagnostic panels. Address any limitations in the study, such as sample size or geographic focus, and suggest future research directions.

Response 4. Agreed the suggestions of reviewer and the suggestions has been duly incorporated in the text.

Comments 5 Concluding : Strengthen the conclusion by emphasizing how the findings contribute to current knowledge and what specific steps should be taken next.

Response 5. Agreed with the reviewer’s suggestion and the points are incorporated in the Text.

  1. Response to Comments on the Quality of English Language

Comments 6 The quality of English does not limit my understanding of the research.

Round 2

Reviewer 1 Report

Comments and Suggestions for Authors

Dear Authors,

All suggestions and considerations were accepted, therefore the manuscript can be accepted in its current form.

Author Response

Thank you very much for the valuable comments and suggestions on this manuscript.

Sincerely

N. Krishnamoorthy

Reviewer 2 Report

Comments and Suggestions for Authors

This MS still needs fundamental revision

32 GoEl – Gr

Fig1 in its current state is redundant because it is non-informative. It could be removed

Fig2. Please, indicate out groups on both trees. Please, indicate 1 and 2 clusters on the tree according to the text of the caption

Fig3. How this tree was constructed? If this is a MP tree, no need to show it, it is enough to say that ML, MP and NJ trees had similar topologies

Please, explain why you think that the blue arrow correctly indicates the ancestor? It is not evident from this tree

275 I do not understand what Figure is it?

Table1 is given in suboptimal way

Fig.4 Please, indicate the same ancestor sequence which was indicated by the blue arrow in Fig. 3 and discuss position of the ancestor in different trees

Table 2. This table needs revision. No need to give the footnotes, they say nothing important in the context of this MS. Why there are empty cells in the Table?

Fig.6 It is not clear why the sequence numbers in the alignment starts with PQ36… but in the Table 2 they starts with PQ38…

Author Response

Thank you very much for the valuable comments and suggestions. We have revised all the suggestions by the reviewer 2, and the response to reviewers' comments is attached herewith.

Thank you.

Sincerely

N. Krishnamoorthy

Reviewer 3 Report

Comments and Suggestions for Authors

The current revised version has been well revised to reflect the points pointed out. Thank you for your hard work.

Author Response

(The authors gave the same response as above.)
